



# Evaluation of Aeolus feature mask and particle extinction coefficient profile products using CALIPSO data

Ping Wang[1], David Patrick Donovan[1], Gerd-Jan van Zadelhoff[1], Jos de Kloe[1], Dorit Huber[2], and Katja Reissig[3]

[1]Royal Netherlands Meteorological Institute (KNMI), Utrechtseweg 297, De Bilt, The Netherlands
[2]DoRIT, 82256 Fürstenfeldbruck, Germany
[3]IB Reissig, Weiheranger 14, 86923 Finning, Germany

**Correspondence:** Ping Wang (ping.wang@knmi.nl)

**Abstract.**

The Atmospheric LAser Doppler INstrument (ALADIN) onboard Aeolus, was the first high-spectral-resolution lidar (HSRL) in space. It was launched in 2018 and re-entered in 2023. The feature mask (A-FM) and extinction profile algorithms (A-PRO) developed for the Earth Cloud, Aerosol and Radiation Explorer (EarthCARE) HSRL Atmospheric Lider (ATLID) have been

adapted to Aeolus, called AEL-FM and AEL-PRO, respectively. These algorithms have been purpose built to process low signal-to-noise ratio space-based lidar signals. A short description of the AEL-FM and AEL-PRO algorithms is provided in this paper. AEL-FM and AEL-PRO prototype products (v1.7) have been evaluated using collocated Cloud-Aerosol Lidar and Infrared Pathfinder Satellite Observations (CALIPSO) Vertical Feature Mask (VFM) product and Level 2 aerosol profile product for two months of data in October 2018 and May 2019. Aeolus and CALIPSO are both polar orbiting satellites but

they have different overpass time. The evaluations are focused on desert dust aerosols over Africa. These types of scenes are often stable in space (tens of km) and time (on the order of 0.5-1 hr), and thus, a useful number of col-located cases can be collected.

We have found that AEL-FM feature mask and the CALIPSO VFM show similar aerosol patterns in the collocated orbits but AEL-FM does not separate aerosol and cloud features. Aeolus and CALIPSO have good agreement for the extinction

coefficients for the dust aerosols, especially for the cloud-free scenes. The Aeolus aerosol optical thickness (AOT) is larger than CALIPSO AOT mainly due to cloud contamination. Because of missing a cross polar channel, it is difficult to distinguish aerosols and thin ice clouds by using the Aeolus extinction coefficients alone.

The AEL-FM and AEL-PRO algorithms have been implemented in the Aeolus level 2A (L2A) processor. The findings here are applicable to the AEL-FM and AEL-PRO products in the L2A Baseline 17. This is the first time the AEL-FM and

AEL-PRO products have been evaluated using CALIPSO data.

## 1 Introduction

The Atmospheric LAser Doppler INstrument (ALADIN) onboard the Aeolus satellite from the European Space Agency was launched in August 2018 and performed measurements for about 5 years. ALADIN is a high spectral resolution lidar (HSRL)





designed to measure wind using Rayleigh and Mie channels at the wavelength of 354.8 nm (Stoffelen et al., 2005; Reitebuch
et al., 2018). Aeolus measured global wind profiles from the ground surface up to about 25 km. The wind products were
assimilated in ECMWF (European Centre for Medium-Range Weather Forecasts) model and proved to be beneficial for the
numerical weather prediction model (Rennie et al., 2020).

Aerosols and clouds can also be measured by Aeolus, however, the lidar is not optimized for aerosol and cloud sensing. For
example, both the range-bins are large (ranging from about 0.25 km to 2 km) and the horizontal resolution is also coarse (3 – 15
km). Despite the limitations of Aeolus for cloud and aerosol remote sensing, valuable information can be extracted and several
algorithms have been developed for the Aeolus level 2A (L2A) aerosol products. The Standard Correct Algorithm (SCA) and
Mie Correct Algorithm (MCA) (Flamant et al., 2008, 2020; Flament et al., 2021) have been implemented in the L2A processor
of the Aeolus mission. The SCA and MCA products are provided for every Aeolus observation interval (about 87 km horizontal
resolution) in 24 altitude bins. The SCA algorithm is very sensitive to noise, while the Aeolus signals had rather low signal
to noise ratio. Ehlers et al. (2022) built upon the SCA to produce a physically constrained Maximum-Likelihood Estimation
(MLE) algorithm. The MLE algorithm has shown considerable noise suppression capabilities compared to the SCA algorithm.

The Earth Cloud, Aerosol and Radiation Explorer (EarthCARE) mission is being implemented by ESA, in cooperation with
the Japan Aerospace Exploration Agency (JAXA), to measure profiles of aerosol, cloud, and precipitation properties together
with radiative fluxes (Illingworth et al., 2015; Wehr et al., 2023). The EarthCARE satellite includes four scientific instruments:
a 94 GHz Doppler cloud profiling radar (CPR), a 355 nm high spectral resolution atmospheric lidar (ATLID), a multispectral
imager (MSI), and a broadband long- and short-wave radiometer (BBR). ATLID is a HSRL depolarization lidar operating at
a wavelength of 355 nm, which measures the co-polar signals, cross-polar signals and Rayleigh signals. Unlike ALADIN,
ATLID does not measure winds and is optimised for aerosol/cloud sensing. The range-gate structure is fixed (100 m from 0 to
20 km and 0.5 km from 20 km to 40 km) and the horizontal resolution is nominally 280 m.

The ATLID Feature Mask (A-FM) product provides a probability mask for the presence of atmospheric features, such as
clouds, aerosols, clear-skies, in the lidar profiles (van Zadelhoff et al., 2023). A-FM is the first processor in EarthCARE L2
processor chain and the feature mask output is used by other processors in the chain (Eisinger et al., 2024). The aerosol
profile retrieval algorithm (A-PRO) uses the feature mask to determine where features are present and to help guide the signal
averaging process. Satellite lidar instruments usually suffer from relatively low signal-to-noise ratios, thus averaging of signals
is needed for the aerosol products, however, the averaging must respect the presence of strong-features. Both A-FM and A-PRO
algorithms have been tested using simulated EarthCARE observations (van Zadelhoff et al., 2023; Donovan et al., 2023, 2024a).

The feature mask algorithm (A-FM) and aerosol profile retrieval algorithm (A-PRO) developed for the ATLID in the Earth-
CARE mission have been adapted to Aeolus, called AEL-FM and AEL-PRO (Donovan et al., 2024a). The AEL-PRO algorithm
takes into account the generally low SNR and uses feature mask as a guideline to average signals to further improve the SNR
in the aerosol profile retrievals.

The Cloud-Aerosol Lidar and Infrared Pathfinder Satellite Observations (CALIPSO) satellite was launched in 2006 to study
the impact of clouds and aerosols on the Earth's radiation budget and climate (Winker et al., 2009) and carried out mea-
surements for 17 years. The Cloud-Aerosol LIdar with Orthogonal Polarization (CALIOP) onboard CALIPSO, is an elastic



backscatter lidar that emits linearly polarized laser light at 532 and 1064 nm and receives both the linear polarized signals and
the cross polarized signals at 532 nm only.

CALIPSO observations were used to characterise the global 3D distributions of aerosols and their seasonal and inter-annual variations (Winker et al., 2013). We have used the vertical feature mask and the aerosol extinction profile data in the Level 2 products, see Sect. 3 for more information.

The AEL-FM and AEL-PRO prototype algorithms v1.7 have been implemented in Aeolus L2A processor but the products
are not available to the public yet. Therefore, we compared the AEL-FM and AEL-PRO prototype products with the CALIPSO products. The AEL-FM and AEL-PRO prototype products were generated using AEL-FM v1.7 and AEL-PRO 1.7.2, using L1B Baseline 14 data. This is the first time the AEL-FM and AEL-PRO products are evaluated using CALIPSO data for dust aerosol scenes.

In this paper we provide a short introduction of the AEL-FM and AEL-PRO algorithms in sect. 2. In Sect. 3, data and method-
ology used in the evaluation of the AEL-FM and AEL-PRO products are described. Sect. 4 presents the results. Conclusions are given in Sect. 5.

## 2  Aeolus feature mask and particle extinction profile retrieval algorithms

The main differences in AEL-FM and AEL-PRO compared to the A-FM, A-PRO algorithms are due to the missing cross polar channel, the coarse and variable vertical bin sizes, the large along-track pixel size, and the large slant viewing angle of 35
degree in the Aeolus measurements. Another difference was the fact that the range-bin settings for Aeolus data were adaptive. That is, they were optimized for wind-retrievals and could change at every "observation" interval. Aeolus data was structured such that each observation interval consisted of a number (e.g. 30) of separate measurements. Each measurement within an observation interval maintained a constant range-gate setting, however, the range-gate settings between each observation often would change. This made horizontal averaging across different observations problematic. These differences meant that the
adaptation of A-FM and A-PRO to Aeolus was far from a trivial task.

Another issue was linked to the need for accurate pure Rayleigh and Mie attenuated backscatter signals (or "cross-talk corrected" signals). To this end, a procedure for producing cross-talk free attenuated backscatter profiles using the Mie Spectrometer (MSP) data alone was implemented. This procedure is briefly outlined in the next section, then AEL-FM and AEL-PRO are briefly described.

## 85  2.1  Attenuated backscatter signals

The AEL-FM and AEL-PRO algorithms use Rayleigh (Ray) and Mie attenuated backscatters (ATBs) derived from the Mie spectrometer (MSP) only. The signals from the Mie spectrometer are imaged onto an Accumulation Charge Coupled Device (ACCD) with 20 detector columns in the spectral domain, where the first and last two columns of the ACCD are used to detect dark currents, the other 16 ACCD columns are used to detect backscatter signals (Reitebuch et al., 2018). The Mie measure-
ments are derived by grouping the CCD pixels close to the Mie peak position together, while the Rayleigh measurements are





derived from the pixels at two sides of the Mie peak. Figure 1 illustrates the peaks in the Mie measurements at the altitude of 11540.66, 10531.65, 9522.71 m (bins 4, 5, 6) at the latitude of 64.74 °N and longitude of -40.55 °E (along track horizontal ground pixel 3) in orbit 646 on 2 October 2018. In this example, the Mie peaks indicate that clouds/aerosols present in bins 5 and 6 but not in bin 4. The MSP data is corrected for dark count and background offset, then corrected for an Effective Mie

Spectrometer Response (EMSR) because the response of each ACCD pixel is different. The EMSR is calculated using cloud-free, aerosol-free Mie measurements per orbit. The EMSR data for orbit 646 is shown in Fig. 2 as an example. The EMSR is normalized so that the mean of the 16 values is 1.0.

Cross-talk between the Mie and Rayleigh signals is accounted for by using pre-calculated cross-talk coefficients. The cross-talk coefficients correspond to a zero Doppler shift and a uniform intensity distribution across the MSP. The EMSR is used

to correct for the non-uniform intensity distribution across the MSP. The cross-talk coefficients are somewhat insensitive to expected Doppler shifts. As a simple way to account for possible Doppler shifts, the centroid of the spectra is calculated and used to adjust the Mie and Rayleigh regions. The absolute calibration coefficients of the ATBs are calculated using simulated Rayleigh ATBs at high altitudes where almost no aerosols and clouds are present. The bins used for calibration are selected using a threshold of scattering ratio (typically 1.1) and the heights of the bins should be above 10 km.

Because the Mie and Rayleigh ATBs are derived from the same MSP, we only need to take into account the parameters related to the MSP in the calibration and cross-talk correction. This results in a system that is easier to quantify than using combined Rayleigh Spectrometer (RSP) and MSP signals. The presence of cross-talk degrades the SNR of the cross-talk corrected ATBs. The cross-talk correction associated with the MSP-only approach degrades the SNR much less than the correction appropriate to using the full MSP and RSP signals. Another advantage of the MSP-only procedure is that the altitude bins of the Rayleigh

and Mie channels can be different but using the Rayleigh signal derived from the MSP solved this problem automatically. Details about the calculations of the ATBs are provided by Donovan et al. (2024b).

## 2.2 Feature mask

The feature mask is an input to the Aeolus aerosol extinction profile retrievals. Because of low signal to noise ratio, the individual measurements have to be averaged before quantitative aerosol retrievals can be performed. However, cloud signals

can be an order of magnitude larger than aerosol signals, so the cloud signals have to be removed when averaging the aerosol signals. A-FM and AEL-FM are described in detail by van Zadelhoff et al. (2023), here a brief overview is given.

The AEL-FM Feature Mask main output is a feature detection probability index ranging between 0 (clear sky) and 10 (likely very thick clouds). The feature detection probability mask is based on exploiting the two-dimensional time-height correlation of the data. AEL-FM detects features using the median-hybrid method (Russ, 2007) (chapter 4) for strong features, and a

data smoothing strategy based on a simplified maximum entropy method (Smith and Grandy, 1985) for the detection of weak features. The advantage of the approach is to enable the retrieval to deal with the low signal to noise ratio at single pixel level.

The feature mask is retrieved at about 3 km (up to 15 km) resolution horizontally. The altitude bin size varies from 0.25 km to 2 km, in total 24 bins, typically from -0.5 km to 20 or 25 km. The configuration of the feature mask algorithm has to be





modified for the Aeolus measurements. The most important configuration parameters are the size of the convolution window
and the maximum signal for the Rayleigh channel.

## 2.3 Aerosol profile retrieval

AEL-PRO is based on the extinction backscatter depolarization (EBD) component of the A-PRO (ATLID Profile) processor
(Donovan et al., 2024a). AEL-PRO is an optimal estimation retrieval algorithm which shares the same lidar forward model as
A-PRO. Figure 3 shows a schematic depiction of the AEL-PRO algorithm.

Similar to A-PRO a two-pass approach is used for retrieving both cloud and aerosol optical properties. Unlike A-PRO, in
AEL-PRO an optimal estimation approach is used for both passes (while in A-PRO the first pass is performed using a direct
retrieval method). In the first pass (Steps 1-5), the retrieval is applied to "strong-feature" screened attenuated backscatter signals
averaged over an ALADIN observation interval. The results of the first pass are treated as an a-priori "background-state" for
the measurement-by-measurement second pass (Steps 6-8). ALADIN did not measure the cross-polarized return signal, such
measurements were not possible for ALADIN. Thus, the classification procedures associated with AEL-PRO are simplified
with respect to those associated with A-PRO (Donovan et al., 2024a; Irbah et al., 2023).

    AEL-PRO employs the principle of Optimal Estimation (Rodgers, 2000) in the retrievals (Steps 5 and 8 in Fig. 3). Like all
optimal-estimation approaches a *cost-function* ($\chi$) is formulated which expresses the sum of the weighted difference between
the observations and the observations predicted by a forward model ($\mathbf{F}$) given a certain state ($\mathbf{x}$) and the weighted difference
between the state and an a priori state ($\mathbf{x_a}$).

    The particular chi-squared cost function used by AEL-PRO can be written as Eq. 1,

$$\chi^2 = [\mathbf{y} - \mathbf{F}(\mathbf{x})]^T \mathbf{S_y^{-1}} [\mathbf{y} - \mathbf{F}(\mathbf{x})] + [\mathbf{x_r} - \mathbf{x_a}]^T \mathbf{S_a^{-1}} [\mathbf{x_r} - \mathbf{x_a}], \tag{1}$$

where

    – $\mathbf{y}$ is the observation vector including the observed Rayleigh and Mie attenuated backscatters.

$$\mathbf{y} = (ATB_{R,1}, ATB_{R,2}, ...ATB_{R,N}, ATB_{M,1}, ATB_{M,2}, ...ATB_{M,N})^T, \tag{2}$$

     where $N$ is the number of range-gates, $ATB_{R,i}$ is the observed Rayleigh co-polar attenuated backscatter and $ATB_{M,i}$
    is the observed attenuated Mie co-polar backscatter.

    – $\mathbf{x}$ is the state-vector defined as:

    $$\mathbf{x} = \log_{10}(\alpha_{M,1}, \alpha_{M,2}, ...\alpha_{M,N}, S_1, S_2, ...S_N, Ra_1, Ra_2, ...Ra_N, C_{lid})^T, \tag{3}$$

where $S$ are the lidar ratios, $R_a$ are the particle effective area radii, and $C_{lid}$ is a factor used to account for calibration
    errors, $\alpha_M$ are the particle extinction coefficients. The log form is used to constrain the retrieved state-vector to be
    positive.





   – $\mathbf{x_a}$ is the a priori state vector. Here defined as a vector consisting of the log base 10 values of the a priori lidar ratios, particle effective area radii and the value of $C_{lid}$ appropriate for calibrated attenuated backscatter signals (i.e. 1). Using a log form here is consistent with the a priori errors being proportional in nature rather than absolute.

$$\mathbf{x_a} = \log_{10}\left(S_{a,1}, S_{a,2}, ...\ S_{a,N}, Ra_{a,1}, Ra_{a,2}, ...Ra_{a,N}, 1\right)^T. \tag{4}$$

   Note that here no a priori constraints are placed upon the log extinction values so that they are not present in the a priori state vector. This leads to the defining of the reduced state vector ($\mathbf{x_r}$) which is just the state-vector excluding the extinction coefficients.

   – $\mathbf{S_a}$ is the a priori error covariance matrix. Here the form of the entries is the one appropriate for a logarithmic state vector i.e.

$$S_{a_{i,i}} = \log_{10}\left(1 + \frac{\sigma^2_{x_{a_i}}}{x_a^2}\right), \tag{5}$$

   where $\sigma_{\mathbf{x_{a_i}}}$ is the a priori (linear) uncertainty assigned to the $i$th component of $\mathbf{x_a}$.

   – $\mathbf{S_y}$ is the observational error covariance matrix. The errors for the Mie and Rayleigh signals at the same altitudes will be correlated due to cross-talk. For details, see Donovan et al. (2024b).

   – $\mathbf{F}$ is the forward model which predicts the Rayleigh and Mie attenuated backscatter profiles given the state vector as an input. The forward model accounts for multiple scattering. The multiple scattering lidar equation used in this work is described in detail in Appendix B of Donovan et al. (2024a) and the exact discrete form used in this work along with its Jacobian is described in Appendix C of Donovan et al. (2024a).

As mentioned earlier, AEL-PRO uses a two-pass approach to cloud-screen and process both strong features (e.g. clouds) and weak features (e.g. aerosols). Pass-I of the algorithm is at the ALADIN so-called observation horizontal resolution of about 90 km while the Pass-II is at the highest available resolution (the measurement scale or about 3 km). The AEL-PRO algorithm first separates strong and weak features using the feature mask as a guidance. The weak features are averaged at horizontal resolution of 90 km (observations). The retrievals are first applied to the weak features. Then the retrievals are run again using the output of pass-I as a priori for every measurement at 3 km horizontal resolution.

A simple classification approach based on the scattering ratio, Mie ATB, and temperature are used to separate water clouds, ice clouds, supercooled water clouds, stratospheric aerosols, stratospheric clouds, and tropospheric aerosols is implemented within AEL-PRO (Step 3 and 6 of Fig. 3). These simple classifications are needed to select a priori values for the state vector and are provided in the output file. The a priori values of lidar ratio (S) and $R_a$ are specified for each category. The a priori values of $\alpha_M$ are computed using calculated scattering ratio, Rayleigh backscatter, and the a priori value of S. A priori values of the state vector for different categories such as aerosols, water clouds, ice clouds, are specified in a configuration file. Where Pass-I produces valid estimates of the lidar ratio, they are used in Pass-II, otherwise the lidar ratio supplied by the classification procedure is used.





The output of AEL-PRO are extinction coefficient, extinction to backscatter ratio, particle effective area radius at middle
altitude of each bin and the variances of these parameters. Example extinction coefficient and lidar ratio retrievals corresponding
to orbit 5221 (2019-07-18) are shown in Fig. 4. Both results from the SCA mid-bin algorithm and AEL-PRO results are
shown. Here data was aggregated to a resolution of 0.5 km (vertically) by 90 km (horizontally). There is a large degree
of correspondence between the SCA and AEL-PRO results, however, the AEL-PRO results are more precise and sensitive,
particularly with regards to the lidar ratio retrievals. In particular, the SCA approach tends to only produce usable estimates
of the lidar ratio for extinction values above 0.05 km$^{-1}$, while AEL-PRO supplies usable estimates of the lidar ratio for
extinctions on the order of 0.002 km$^{-1}$. The more precise nature of the AEL-PRO results can again be seen in Fig. 5. Here
the difference in precision (noise) is evident between the SCA and AEL-PRO results. This difference in precision is due to the
combined effects of both more precise attenuated backscatter profiles estimates as discussed earlier and the regularization (or
stabilization) effect afforded by the optimal estimation approach used by AEL-PRO. It can also be seen that the resolution of
the AEL-PRO products is finer than the SCA products at lower altitudes. This is a direct consequence of the need to create a
merged grid to combine the MSP and RSP signals used by the SCA process. The AEL-PRO approach uses the MSP vertical
grid which tends to have a finer resolution than the RSP vertical grid.

## 3 Data and Methodology

### 3.1 CALIPSO data

CALIPSO v4.51 L2 aerosol profile product (CAL_LID_L2_05kmAPro-Standard-V4-51) at a uniform spatial resolution of
5 km horizontally and vertical 60 m from altitude range of 30 km to -0.5 km has been used in this analysis. Although the aerosol
data is provided at 5 km horizontal grid, the products may have been averaged up to 80 km horizontally before being detected
by the CALIPSO feature finder algorithm. The aerosol profile product reports profiles of particle extinction and backscatter,
additional profile information and layer optical depths. These products are produced using the same basic algorithm (Young
and Vaughan, 2009; Young et al., 2018).

In order to compare with Aeolus extinction coefficient profiles, we used the CALIPSO tropospheric aerosol extinction profile
between 20 km and 0.5 km. The CALIPSO optical thickness (AOT, Column_Optical_Depth_Tropospheric_Aerosols_532) and
extinction coefficient at 532 nm (Extinction_Coefficient_532 ) were converted to AOT and extinction coefficient at 355 nm.
The Angström coefficient of 0.55 for dust was used to convert the AOT and extinction coefficient from 532 to 355 nm (Amiridis
et al., 2015). We also used CALIPSO L2 Vertical Feature Mask (VFM) images (Vaughan et al., 2009), which describes the
vertical and horizontal distribution of cloud and aerosol layers observed by the CALIOP lidar.

### 3.2 Aeolus data

During the Aeolus mission, switches between operation using the primary (FM-A) to the secondary laser (FM-B) were made.
Data from September 2018 to June 2019 was measured by the laser FM-A, data from July 2019 to November 2022 was



measured by the laser FM-B, data from December 2022 to April 2023 was measured by FM-A again. We used the Aeolus L1B
      products from FM-A reprocessed data Baseline 14 (reprocess v3) from September 2018 to June 2019. We processed the AEL-
      FM and AEL-PRO data using the prototype codes at KNMI (v1.7) using reprocessed L1B data Baseline 14. The prototype
      version v1.7 has been implemented in the L2A processor but the Aeolus L2A data has not yet been reprocessed using v1.7.
      We have, however, verified the implementation of AEL-FM and AEL-PRO for several versions in the L2A processor using
the prototype codes and found that the AEL-FM and AEL-PRO in L2A are almost identical to the prototype products (see
      Appendix A).

      The temperature and pressure data was taken from the L2A product, which was originally supplied by the ECMWF (Rennie
      et al., 2020). Aeolus had 15 orbits per day with orbit repeating cycle of 7 days. Typically one L1B file has one full orbit of data.
      The AEL-FM and AEL-PRO products are provided for every measurements at the Mie measurement altitude bins, therefore
the horizontal resolution is about 3 km (up to 15 km) and vertical resolution varies from 0.25 to 2 km. The AEL-FM and AEL-
      PRO products include Rayleigh ATB, Mie ATB, feature mask, EMSR, particle extinction profile, extinction to backscatter ratio
      profile, particle effective area radius profile, simple classification, and additional information used in the retrievals.

### 3.3   Data collocation

      Collocated Aeolus and CALIPSO data was selected per orbit between the longitude range of [60 °W, 60 °E] and the latitude
range of [60 °S, 60 °N]. The time differences between collocated data were smaller than 4 hours. The collocated orbits were
      selected with longitude differences of smaller than 3 degrees. Then, the collocated pixels closest in latitude were selected.
      Quality flags in the CALIPSO and Aeolus data were used to select the most reliable data. The CALIPSO data with an Extinc-
      tion_QC_Flag_532 value of 0 or 1 (Young et al., 2018) and Aeolus data with a quality index smaller than 32 were used. In the
      comparison with the Aeolus data, the CALIPSO L2 aerosol extinction profiles were averaged using the vertical bins of Aeolus.
The aerosol optical thickness of AEL-PRO was integrated using the extinction profile from the surface to 6 km. The reason
      for this choice is explained later in Section 4.1.2. We did not compare the backscatter profiles between Aeolus and CALIPSO
      because Aeolus measured only co-polar signal of circular polarization and CALIPSO measured linear polarization, therefore
      the backscatter profiles are not directly comparable.

      Figure 6 illustrates the collocated Aeolus orbit 646 on 2 October 2018 and orbit 766 on 10 October 2018 with CALIPSO
orbits in the selected region. In the southern hemisphere part of orbit 646, the Aeolus and CALIPSO orbits have large difference
      in longitude but in the northern hemisphere part, these two orbits satisfy the selection criteria. So in the quantitative evaluations,
      we only selected the data in the north part of the orbits. In the images, we show the orbits from 60 °S to 60 °N.





## 4 Results

### 4.1 Comparison of Aeolus and CALIPSO products per orbit

#### 4.1.1 Feature masks

The AEL-FM data has been compared with the CALIPSO VFM by van Zadelhoff et al. (2023) for dust plumes in cloud-free scenes with almost no cloud above the dust plumes. There are often thin (cirrus) clouds above dust plumes or clouds and aerosols in the boundary layers.

Figure 7 shows the feature mask of CALIPSO for the orbit between 11:09 and 11:36 UT on 2 October 2018 and the collocated
Aeolus orbit 646 between 15:41 and 16:06 UT, the geolocation of this orbit is shown in Fig. 6. In Fig. 7a, the CALIPSO feature mask images (separated in 2 panels) taken from the CALIPSO online browser archive are shown. The second granule has good collocation with the Aeolus orbit 646. CALIPSO detects thick high clouds with a cloud top height between 10 and 12 km at about 40 – 34 °S, the thick clouds are also detected in Aeolus feature mask in this latitude range. The cloud top height is about 12–13 km. The real cloud base is not detected since the CALIPSO and Aeolus signals are both fully attenuated. Although there
are 4.5 hours time difference and about 5 degree longitude difference, CALIPSO and Aeolus detect similar cloud features. Along the orbit further to the north, both CALIPSO and Aeolus detect aerosols above ocean up to 1 km. However, CALIPSO also detects some boundary layer clouds on top of the aerosols at 1 km, Aeolus does not show a cloud layer above the aerosol layer, mostly due to the coarse vertical bins. In this orbit, the Aeolus vertical bin size is 250 m below 2 km, and from 2 to 12.5 km the bin size is 1 km. In the CALIPSO feature masks between 20 and 2 °S, there are clouds at 5 km and aerosols below
the clouds. These features are also detected by Aeolus. In the second granule of CALIPSO, the aerosol layer between 10–19 °N is detected from the surface up to 5 km, with a few clouds at the top of the aerosol layer, similar features are detected by AEL-FM. North of 27 °N, high clouds at about 12 km are observed in the Aeolus data but in the CALIPSO data the high clouds appear at north of 33 °N. Consequently, between the latitude range of 27.5 and 40 °N some aerosols close to the surface are not detected in the Aeolus data due to the clouds.

Figure 8 shows the comparison of Aeolus and CALIPSO feature mask products on 10 October 2018, with the orbit depicted in Fig. 6(b). CALIPSO measured dusts in 37–12 °N and biomass burning aerosols in 5.54 – 24 °S on 10 October 2018. The Aeolus feature mask exhibits a similar shape and altitude compared to the CALIPSO feature mask for both clouds and aerosols. The aerosol plumes are from the ground surface up to 5 km, with some small scattered clouds on top of the aerosols. The heights of the aerosol plumes are higher than the aerosol plumes in Fig. 7. The low clouds at about 1 km over ocean, for
example between 48.13 – 59.95 °S, are also detected as thick cloud in the Aeolus feature mask. However, Aeolus feature mask does not provide aerosol types.

In general, AEL-FM product compares well with the CALIPSO features, but due to the time differences and large vertical bin sizes, we cannot expect the same features everywhere in the orbits. The cloud features often have some differences in height and geolocations. So in the evaluation of the extinction profiles, we only compared the aerosol extinction profiles between
Aeolus and CALIPSO.





### 4.1.2 Extinction coefficient profiles

The particle extinction coefficient profiles from CALIPSO and Aeolus for the collocated orbits on 2 October 2018 are shown in Fig. 9. The aerosol extinction profiles from AEL-PRO were selected based on the AEL-PRO classification data (index = 103 for aerosols). The aerosol extinction image of CALIPSO looks cleaner than the extinction image of Aeolus. For the aerosols below 5 km, the two images have similar pattern and colors. So qualitatively, the CALIPSO and Aeolus aerosol extinction profiles are comparable. The Aeolus extinction profiles exhibit more low values in the range of $10^{-5}$ to $10^{-6}$ m$^{-1}$. The large extinction coefficients close to the tropopause are most likely from clouds, not aerosols. Because there is no cross polar Mie channel in Aeolus, it is difficult to distinguish thin ice clouds from aerosols (much of the return from ice clouds is depolarized and thus not detected by Aeolus, this results in a lower apparent backscatter which decreases the contrast between less depolarizing elevated aerosols). The classification of cloud and aerosol is mainly based on the threshold applied to the backscatter coefficient. The images are plotted at 3 km horizontal resolution, but the actual resolution is roughly 90 km because of horizontal averaging. As shown in Fig. 9b, there are some high resolution horizontal pixels but more horizontally averaged pixels. The extinction-to-backscatter (lidar) ratio profiles are also shown in 9d. The lidar ratio image looks more noisy than the extinction image because it is the ratio between two small values. We do not compare the lidar ratio with CALIPSO data in this paper.

Figure 10 shows the comparison of aerosol extinction profiles between CALIPSO and Aeolus for orbit 766 on 10 October 2018. Figure 10(b) shows all Aeolus extinction profiles, both clouds and aerosols, where there are lots of extinction coefficients greater than $10^{-3}$ m$^{-1}$. The total extinction profiles include the aerosol extinction profiles which are very similar to the CALIPSO aerosol extinction profiles. After the cloud contaminated bins are removed, the Aeolus aerosol extinction profiles have similar colors to the CALIPSO extinction profiles. However, it clearly indicates that too many bins at the top of the aerosol plumes between 2.5 km and 5 km are removed from the Aeolus extinction profiles. The aerosol extinction coefficients between 5 km and the tropopause height are also too large compared to the CALIPSO aerosol extinction profiles. We do not find difference between the lidar ratios for the dust and biomass burning aerosols in the Aeolus data. The distribution of the lidar ratio between 20 and 150 sr for the orbit 766 is shown in Fig. 11. A Gaussian function is fitted to get the center of the distribution at about 63 sr. A similar lidar ratio value was employed by Song et al. (2023) in their evaluation of the Aeolus L2A SCA middle bin aerosol product with CALIPSO. Song et al. (2023) found that using the dust lidar ratio of 63.5 sr derived from collocated Aeolus data (L2A SCA middle bin), aerosol optical thickness values retrieved from CALIOP were increased by 46%, which improved the comparison with MODIS data. This suggests that the AEL-PRO retrieved lidar ratios are reasonable for the aerosol scenes.

To provide further insight into the differences between the Aeolus and CALIPSO extinction profiles, we selected three collocated extinction profiles at latitudes of 30 °N, 15 °N, and -9.5 °N. The longitude differences of the CALIPSO and Aeolus orbits are within 1.0 degree. The profiles were selected at cloud-free regions according to the extinction images. The CALIPSO extinction coefficients were interpolated at the middle of each Aeolus altitude bin. The errors of the CALIPSO extinction coefficients were also interpolated at the Aeolus altitude bins. Figure 12 shows these three collocated extinction profiles, which is a zoomed view of the extinction profile images. The Aeolus extinction profiles are the full profiles, clouds are not removed.



We can see that the extinction profiles have good agreement if the clouds can be removed properly. Although some bins have clouds, it seems that the retrieved aerosol extinction profiles are not affected by the clouds in other bins.

## 4.2 Comparison of monthly data

### 4.2.1 Aerosol extinction coefficient profiles

We processed the AEL-FM and AEL-PRO data in September and October 2018, May and June 2019 because there were more
aerosol events in the Saharan deserts in summer months. Unfortunately there was missing data in CALIPSO in September 2018 and only half month of Aeolus data was available in June 2019. So for the statistics we use the collocated data about 48 orbits in October 2018 and 43 orbits in May 2019. The Aeolus and CALIPSO extinction profiles are further selected for the region within the longitude range of -10 to 50 °E, the latitude range of 0 - 30 °N, because the aerosol types are mainly dusts in this region. The AEL-PRO aerosol extinction coefficient in each Aeolus altitude bin was compared with the CALIPSO aerosol
extinction that was averaged in the Aeolus altitude bin and extrapolated to 355 nm.

The scatter plot of Aeolus and CALIPSO aerosol extinction coefficients for this region is shown in Fig. 13. The aerosol extinction coefficients show reasonable agreement between Aeolus and CALIPSO for October 2018 and May 2019. The data has large scatter which can be explained by the time differences and the possible evolution of aerosols. When selecting collocated extinction coefficients, some Aeolus aerosol extinction coefficients affected by clouds may be removed due to the absence of
CALIPSO aerosol extinction data in these bins.

Figures 14 shows the monthly mean extinction profiles, same data as in Fig. 13 but averaged in vertical bins. Both profiles show that the dust layer is mainly below 5 km, the extinction coefficient is larger at the ground surface and decreases at higher altitude until 5 km. AEL-PRO extinction coefficient estimates are larger than CALIPSO close to the ground surface and above 7.5 km. It indicates the possible impact of clouds in the AEL-PRO aerosol profiles. It is also known that CALIPSO
underestimates the aerosol extinction coefficients in the free troposphere due to limited sensitivity (Winker et al., 2013). In fact the AEL-PRO results seem more consistent with e.g. results from airborne lidar (Winker et al., 2013).

### 4.2.2 Aerosol optical thickness

The Aeolus AOT was integrated using the aerosol extinction profile from the ground surface to 6 km, but not the whole profile. As can be seen in the Aeolus aerosol extinction image, AEL-PRO often classified thin clouds as aerosols. In these two months,
there were very few aerosols above 6 km based on the CALIPSO images, so we only used extinction profiles until 6 km to calculate the AOT. The CALIPSO AOT values at 532 nm were taken from the L2 product, then they were converted to the AOT at 355 nm.

Figures 15 and 16 show the AOT maps for October 2018 and May 2019 derived using AEL-PRO extinction profiles and the CALIPSO AOT, respectively. The CALIPSO AOT map shows clearly more aerosols over land and the transport of Saharan
dust over ocean, with the AOT values mostly larger than 0.3. In the Aeolus AOT map, over lands, the values are similar to the CALIPSO AOT. However, over ocean where CALIPSO shows lower AOT, Aeolus often shows high AOT. This suggests



**Table 1.** Monthly mean AOT and extinction coefficient (EXT) for CALIPSO and Aeolus in selected region for dust aerosols.

| Date | Parameter | CALIPSO mean (std) | Aeolus mean (std) | corr. coef. | Number of data |
|------|-----------|--------------------|--------------------|-------------|----------------|
| October 2018 | AOT all | 0.415 (-) | 0.765 (-) | - | - |
| | AOT <1 | 0.306 (0.197) | 0.314 (0.169) | 0.138 | 13677 |
| | EXT all | $1.498 \times 10^{-4}$ (-) | $4.726 \times 10^{-4}$ (-) | - | - |
| | EXT $[1 \times 10^{-5}, 1 \times 10^{-3}]$ m$^{-1}$ | $1.207 \times 10^{-4}$ ($1.184 \times 10^{-4}$) | $1.250 \times 10^{-4}$ ($1.321 \times 10^{-4}$) | 0.200 | 52762 |
| May 2019 | AOT all | 0.565 (-) | 0.832 (-) | - | - |
| | AOT < 1 | 0.311 (0.241) | 0.318 (0.185) | 0.246 | 13761 |
| | EXT all | $1.732 \times 10^{-4}$ (-) | $3.919 \times 10^{-4}$ (-) | - | - |
| | EXT $[1 \times 10^{-5}, 1 \times 10^{-3}]$ m$^{-1}$ | $1.237 \times 10^{-4}$ ($1.144 \times 10^{-4}$) | $1.215 \times 10^{-4}$ ($1.343 \times 10^{-4}$) | 0.257 | 44237 |

that some boundary layer clouds or cirrus were included in the Aeolus AOT calculations. Because of no depolarization data, AEL-PRO cannot really distinguish thin clouds and aerosols. The coarse vertical bin size may result in clouds and aerosols being in the same vertical bin.

The Aeolus and CALIPSO AOT values are further selected for the region within the longitude range of -10 to 50 °E, the latitude range of 0 - 30 °N, because the aerosol types are mainly dusts and there are relatively high AOT values in this region. The scatter plot of Aeolus and CALIPSO AOT at 355 nm for this region is shown in Fig. 17. The Aeolus and CALIPSO AOTs have some correlations. The agreement is better at the small AOT than at the large AOT. It is possible that the small AOT represents background dust aerosols, which is more stable than the dust plumes with large AOT. If the AOT values larger

than 1 are excluded, the mean Aeolus AOT is 0.314 ($\pm$0.169) in October 2018, 0.318 ($\pm$0.185) in May 2019, and the mean CALIPSO AOT is 0.306 ($\pm$0.197) in October 2018 and 0.311 ($\pm$0.241) in May 2019. The results are summarised in Table 1.

## 5   Conclusions

The Aeolus AEL-FM feature mask and AEL-PRO extinction profile products have been compared to the CALIPSO vertical feature mask and extinction profile products for desert dust aerosols over Africa using two months of collocated data in

October 2018 and May 2019. Generally, Aeolus feature masks appear at similar altitude locations as the CALIPSO vertical feature masks although there are 4 hours time differences. The extinction profiles have good agreement for cloud-free aerosol extinction profiles. The monthly mean extinction coefficients are about $1.2 \times 10^{-4}$ m$^{-1}$, similar for Aeolus and CALIPSO if we limit the individual values between $1.0 \times 10^{-5}$ and $1.0 \times 10^{-3}$ m$^{-1}$. Without this limitation, the Aeolus monthly mean extinction coefficients are about 2-3 times larger than the CALIPSO extinction coefficients, which suggests the contamination

by thin clouds. The monthly mean AOT values are also similar between Aeolus and CALIPSO if the individual AOT values are limited to between $1.0 \times 10^{-6}$ and 1. Without the limit of the AOT, the Aeolus monthly mean AOT values are 1.5 to 1.9 times larger than the CALIPSO AOT values in the selected dust aerosol region.



The separation of aerosols from clouds has to be improved in the Aeolus aerosol products. However, due to the missing cross polar channel in the Aeolus measurements, consequently no depolarization product, it is difficult to separate aerosols and thin clouds. Also because of the large vertical bins and horizontal pixels, some bins may partly filled with clouds. We hope the cross polar channel can be included in Aeolus-2 to provide better aerosol and cloud identification.

The AEL-FM and AEL-PRO products have been implemented and verified in Aeolus Baseline 16 L2A product. We expect the AEL-FM and AEL-PRO products in the L2A Baseline 16 and later baselines have a similar quality as the prototype products used in this analysis.

In spite of the difficulties in separating aerosol and cloud in the Aeolus observations, the agreement between Aeolus feature mask, extinction coefficient profile with CALIPSO L2 products gives us confidence in the A-FM and A-PRO products to be produced by EarthCARE. It is expected that the addition of a depolarization channel, better SNR, and better resolution for ATLID will result in an improved ability to separate clouds and aerosols.

*Data availability.* L1B Baseline 14 data are available from the Aeolus Data Dissemination Facility http://aeolus-ds.eo.esa.int/. AEL-FM, AEL-PRO data used in the paper is available from the authors. AEL-FM, AEL-PRO data is available in Aeolus L2A from Baseline 16.



**Appendix A: Verification of AEL-FM and AEL-PRO in Aeolus L2A products**

The AEL-FM and AEL-PRO products in the L2A were verified with the prototype products during the implementation in L2A processor from versions 3.14 to 3.17 (Baselines 14 to 17). The products were also verified during the 3rd and 4th data reprocessing. We found that the L2A and prototype products are almost identical for most orbits. Figures A1 and A2 show an
example of the verification of the prototype and L2A AEL-FM and AEL-PRO products for L2A version 3.16.4 (Baseline 16). Figure A3 shows a scatterplot of the L2A AEL-PRO extinction coefficients versus the prototype. There are only a few data points having large differences between the L2A and the prototype extinction coefficients.

*Author contributions.* PW analysed the data and wrote the main part of the manuscript, GJZ and DD developed the AEL-FM, AEL-PRO prototype codes. JK contributed to the Aeolus L1B and L2A data processing. DH and KR implemented the prototype codes in the Aeolus
L2A processor. All authors contributed to the writing and editing of the manuscript.

*Competing interests.* The authors declare that they have no conflict of interest.

*Acknowledgements.* This work has been funded by the European Space Agency (ESA) in the framework of the activities of the Aeolus Data Innovation and Science Cluster (DISC) consortium. The CALIPSO images were taken from https://www-calipso.larc.nasa.gov/products/lidar/brows_images/production/. The CALIPSO data were ordered from https://www-calipso.larc.nasa.gov/products/.



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





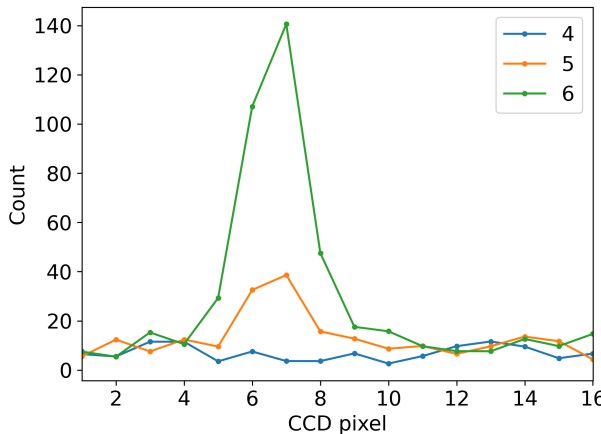

**Figure 1.** Mie measurements at bins 4, 5, 6 in orbit 646 on 2018-10-02 cross track pixel 3, height bins 3-6.

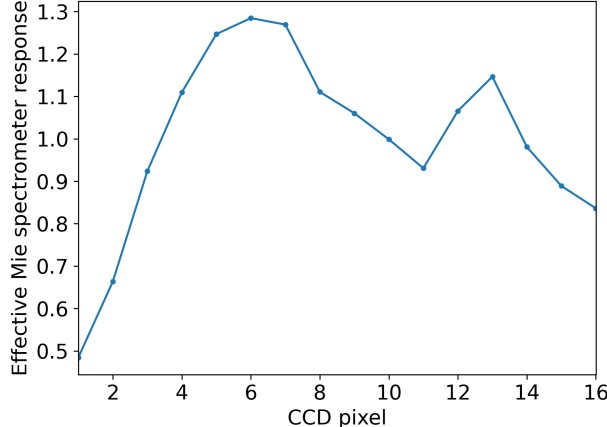

**Figure 2.** Effective Mie spectrometer response (EMSR) for orbit 646 on 2 October 2018.



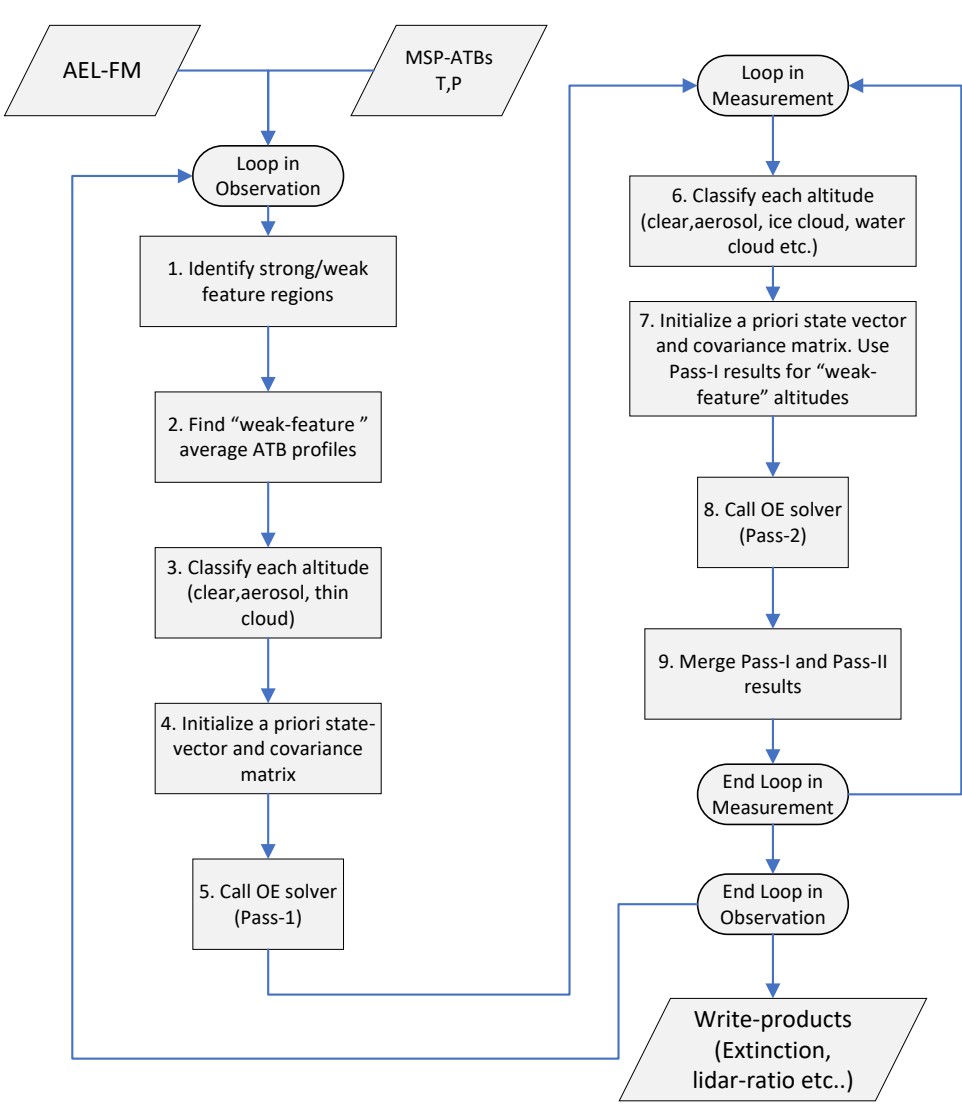

**Figure 3.** Schematic depiction of the AEL-PRO optimal estimation retrieval algorithm.





**Figure 4.** SCA mid and AEL-PRO (both baseline 2A16) retrievals of (a, b) particulate extinction coefficient and (c, d) lidar ratio for orbit 5221 (2019-07-18). The lower thick black line represents the surface heights, the back contour lines represent the temperatures, and the magenta symbols represent the tropopause heights.







**Figure 5.** AEL-PRO (a, b, black) and SCA mid (c, d, red) profiles of retrieved extinction coefficient and lidar ratio (with errorbar) for observation 51 (approximately 76$^{o}$N, 201$^{o}$E) for orbit 5221. The blue line is the temperature profile (upper x-axis scale).



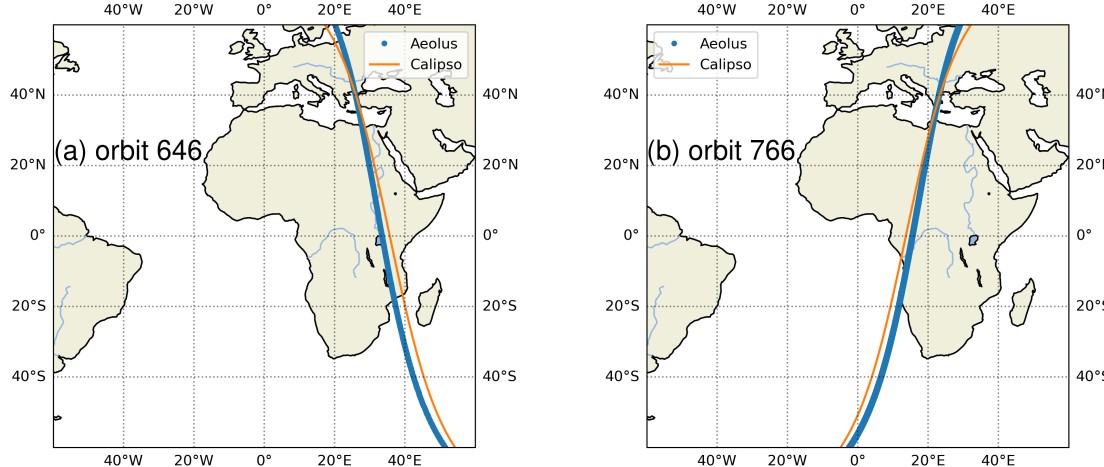

**Figure 6.** Aeolus (a) orbit 646 on 2 October 2018 and (b) orbit 766 on 10 October 2018 with collocated CALIPSO orbits at 10:56:23 UT on 2 October (daytime orbit) and at 00:27:07 UT on 10 October 2018 (nighttime orbit), respectively.





**Figure 7.** Comparison of Aeolus orbit 646 on 2 October 2018 and collocated CALIPSO orbit for feature masks, (a) CALIPSO vertical feature mask, (b) Aeolus feature mask, (c) Aeolus Mie attenuated backscatter.





**Figure 8.** Comparison of Aeolus orbit 766 on 10 October 2018 and collocated CALIPSO orbit for feature masks, (a) CALIPSO vertical feature mask, (b) CALIPSO aerosol subtypes, (c) Aeolus feature mask, (d) Aeolus Mie attenuated backscatter.





**Figure 9.** Comparison of Aeolus orbit 646 on 2 October 2018 and collocated CALIPSO orbit for extinction coefficient profiles, (a) CALIPSO tropospheric aerosol extinction coefficient profiles (L2 5 km aerosol profiles v4.51), (b) Aeolus extinction coefficient profiles for all measurements, (c) Aeolus extinction coefficient profiles for tropospheric aerosols, (d) Aeolus lidar ratios for tropospheric aerosols with extinction coefficients greater than $1.0 \times 10^{-5}$ m$^{-1}$. The purple symbols mark the tropopause heights in km. The thin black contours indicate the atmospheric temperatures (unit K). The thick black line represents the surface heights (unit km).



**Figure 10.** Comparison of Aeolus orbit 766 on 10 October 2018 and collocated CALIPSO orbit for extinction coefficient profiles, (a) CALIPSO tropospheric aerosol extinction coefficient profiles (L2 5 km aerosol profiles v4.51), (b) Aeolus extinction coefficient profiles for all measurements, (c) Aeolus extinction coefficient profiles for tropospheric aerosols, (d) Aeolus lidar ratios for tropospheric aerosols with extinction coefficients greater than $1.0 \times 10^{-5}$ m$^{-1}$. The purple symbols mark the tropopause heights in km. The thin black contours indicate the atmospheric temperatures (unit K). The thick black line represents the surface heights (unit km).





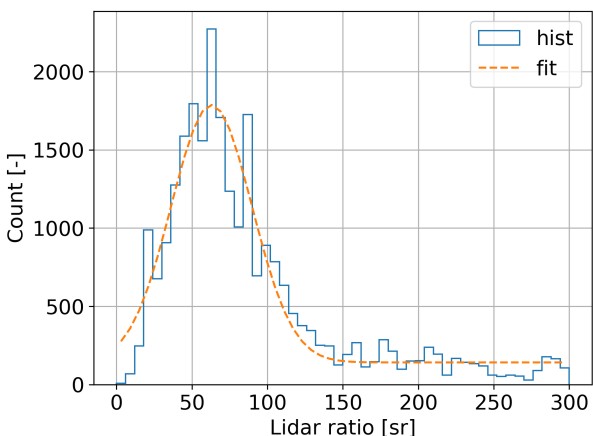

**Figure 11.** Histogram of Aeolus lidar ratios for orbit 766 on 10 October 2018.

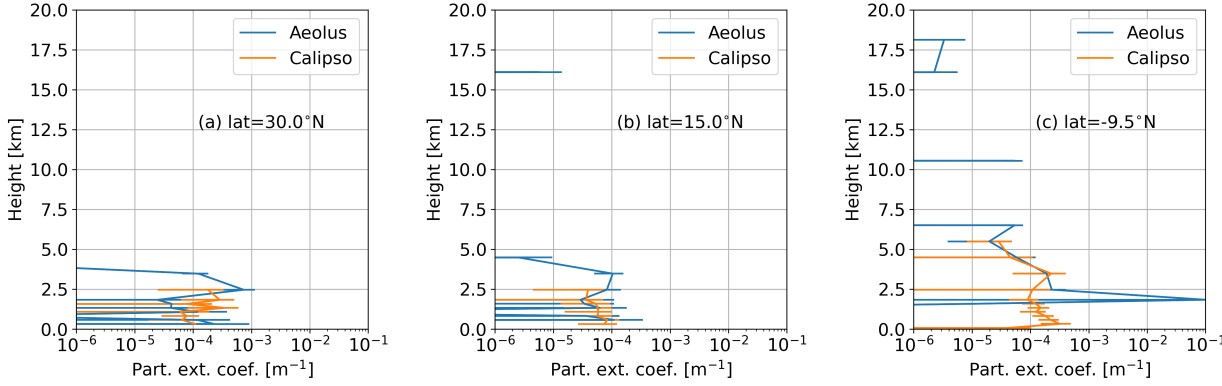

**Figure 12.** Comparison of Aeolus and CALIPSO extinction coefficient profiles at 3 locations in Aeolus orbit 766 on 10 October 2018 at latitudes (a) 30 °N, (b) 15 °N, (c) -9.5 °N, respectively, same data as Fig. 10(a, b).





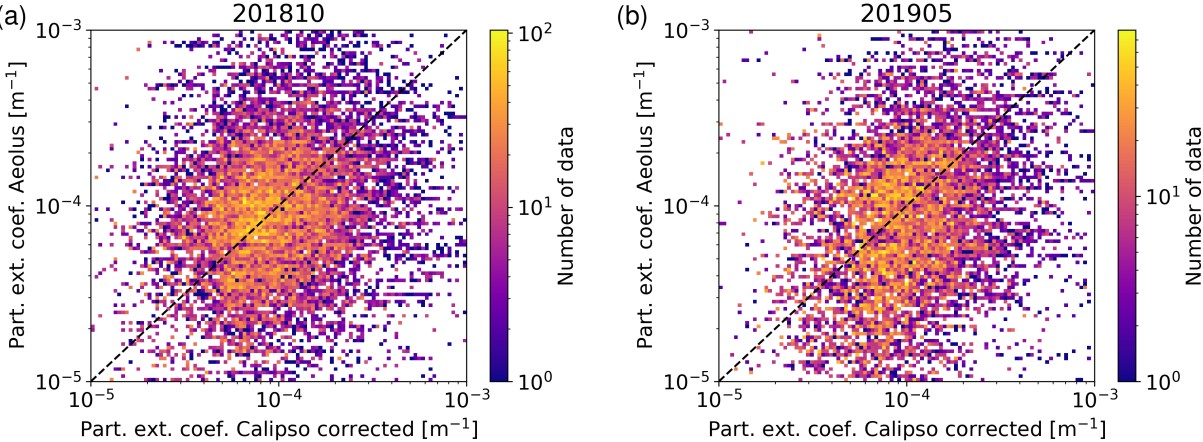

**Figure 13.** Scatter plots of Aeolus tropospheric aerosol extinction coefficients versus CALIPSO tropospheric aerosol extinction coefficients for (a) October 2018 and (b) May 2019, in the region of longitude -10 to 50 °E, latitude 0 to 30 °N. The colorbar indicates the number of data points on a logarithmic scale.

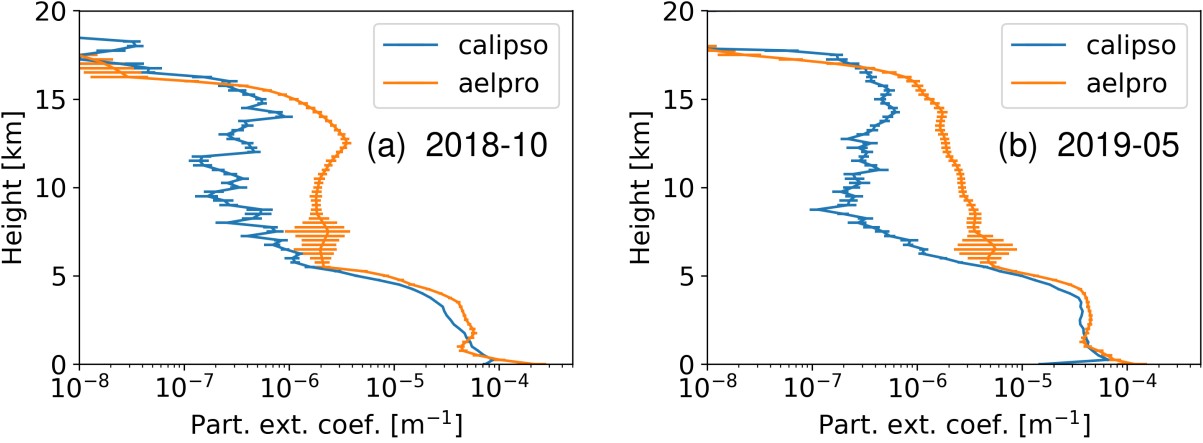

**Figure 14.** Monthly mean CALIPSO and Aeolus extinction coefficient profiles for collocated orbits, (a) October 2018, (b) May 2019, in the region of longitude -10 to 50 °E, latitude 0 to 30 °N. Extinction coefficient profiles with AOT > 1 are excluded in both CALIPSO and Aeolus extinction coefficient profiles.





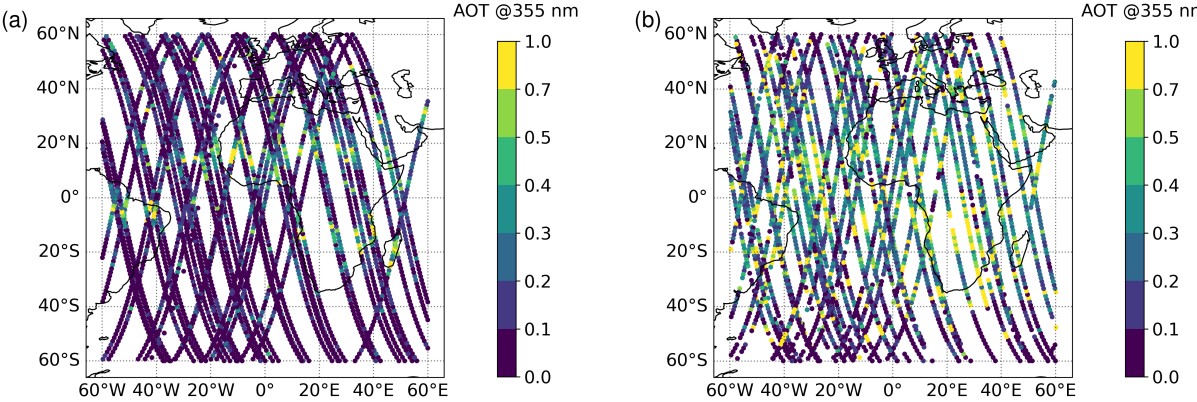

**Figure 15.** Maps of (a) CALIPSO and (b) Aeolus tropospheric aerosol optical thickness in October 2018.

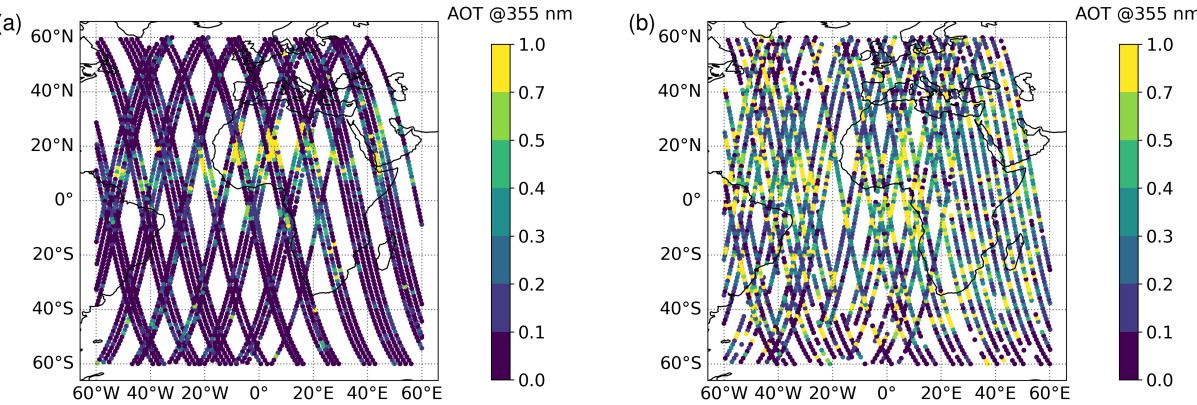

**Figure 16.** Maps of (a) CALIPSO and (b) Aeolus tropospheric aerosol optical thickness in May 2019.



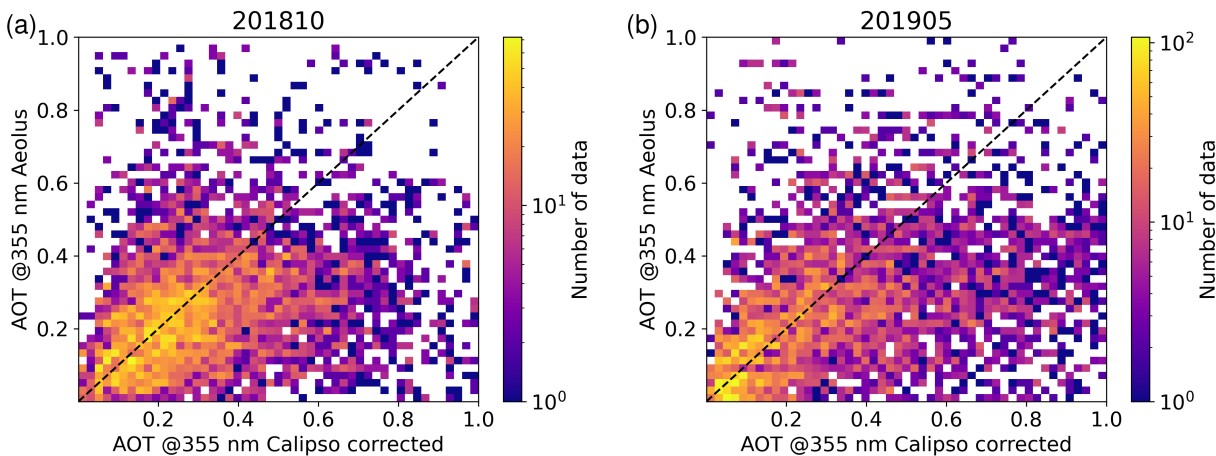

**Figure 17.** Scatter plots of Aeolus tropospheric aerosol optical thickness versus CALIPSO tropospheric aerosol optical thickness for (a) October 2018 and (b) May 2019, in the region of longitude -10 to 50 °E, latitude 0 to 30 °N. The colorbar indicates the number of data points on a logarithmic scale.





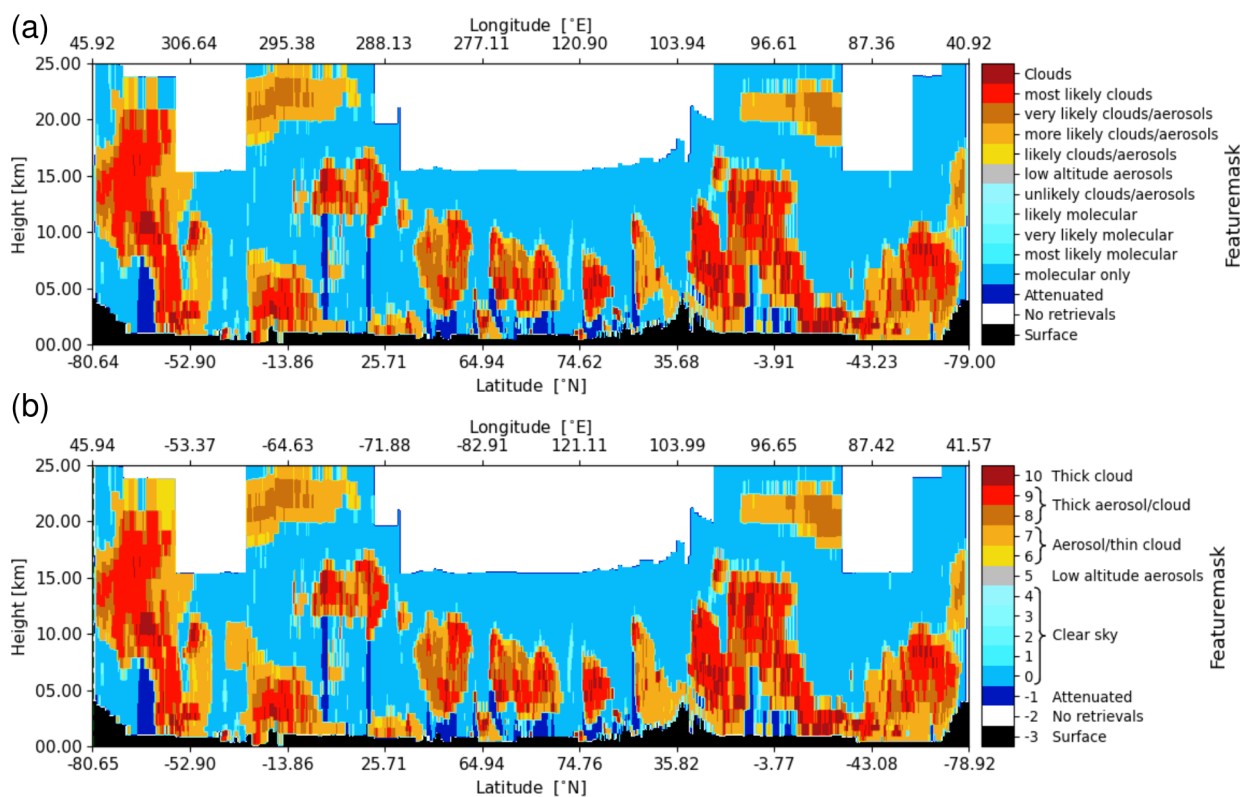

**Figure A1.** Comparison of the AEL-FM feature mask in (a) L2A with (b) the prototype product for orbit 23453 on 9 September 2022. L2A version 3.16.4 (Baseline 16).





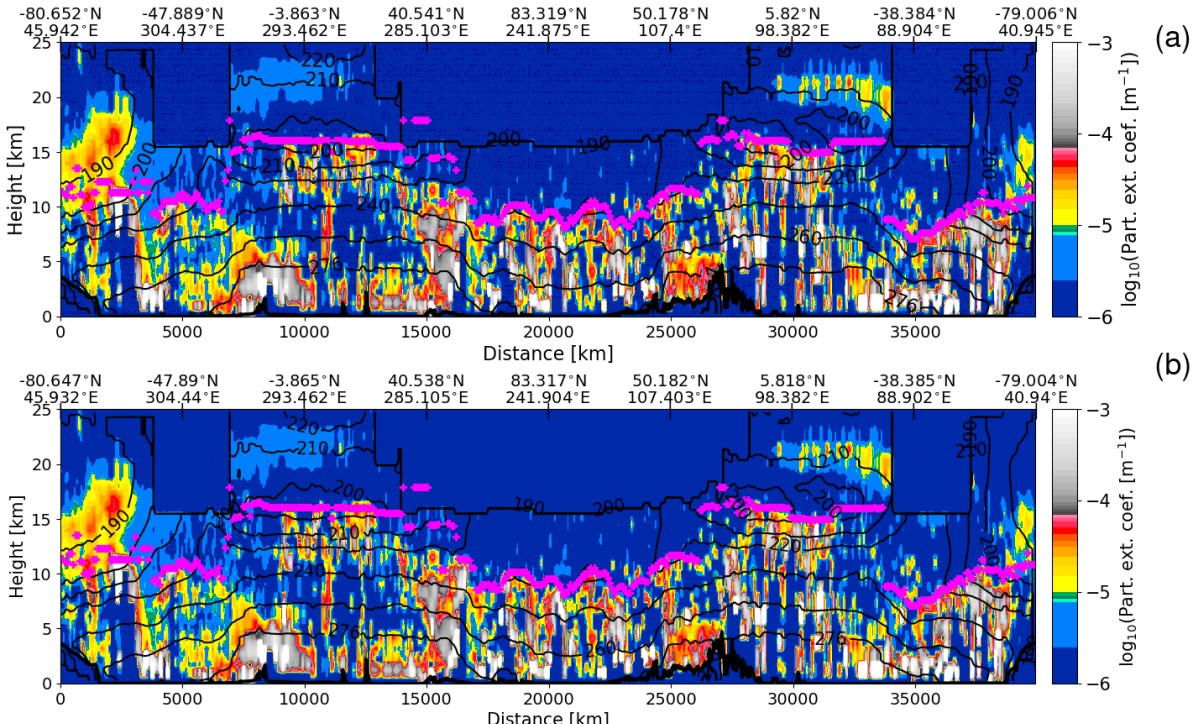

**Figure A2.** Comparison of AEL-PRO extinction coefficient profiles in (a) L2A and (b) the prototype for orbit 23453 on 9 September 2022. L2A version 3.16.4 (Baseline 16). The purple symbols mark the tropopause heights in km. The thin black contours indicate the atmospheric temperatures (unit K). The thick black line represents the surface heights (unit km).

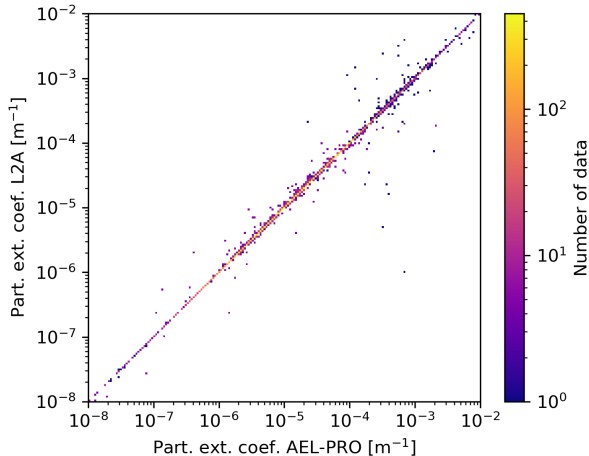

**Figure A3.** Scatterplot of AEL-PRO extinction coefficients in L2A versus the prototype for orbit 23453 on 9 September 2022, same data as in Fig. A2. The colorbar indicates the number of data points on a logarithmic scale.