# Peer review of "Evaluation of Aeolus feature mask and particle extinction coefficient profile products using CALIPSO data"

_EGUsphere, 2024_

## Author Comment (AC1)

We appreciate referee #1 for the comments and suggestions. We have answered all questions and revised the manuscript. We also modified figures 4, 14, 15, 16, so that the AEL-PRO figures come first, then the SCA-mid or CALIPSO figures. In this document, the answers are given in blue text.

The manuscript presents and discusses an algorithm that is used to analyze aerosol-related observations with ESA's spaceborne wind Doppler lidar AEOLUS. The unique methodology is originally designed to analyze EarthCARE space lidar observations (feature mask algorithm, aerosol profile algorithm).

The respective AEOLUS aerosol products are compared with observations performed with NASA's CALIPSO space lidar.

I clearly recommend publication.

I have only minor points

P5, L149: in Eq.3 we have … Ra … then in L150 we have … R_a …. please harmonize!

We have changed 'R_a' to 'Ra'.

P5, L156: In Eq.4 we have Ra_a,1 and so on. All this is a bit confusing!

Ra for effect area radius. '_a' refers to the a priori. Sorry for so many 'a' here.

P7, L197: To my opinion, a discussion of the findings in Figures 4 and 5 is missing! Please, explain the observed features, at least a bit! The paper should not only focus on technical and data analysis points.

For example, the feature in Figure 4 (0-12.5km, and 15-20.0km , -63.2N, 228.3E, winter, July 2019) is interesting, I mean this column-like red/yellow feature? What is it? Smoke? Volcanic aerosol at 15-17.5 km height? The lidar ratio seems to be around 50 sr at 355 nm. So that could have been smoke! The particles were probably spherical so that AEOLUS data can provide reliable lidar ratios.

The lidar ratios in Figure 4 are often in yellow (around 100sr). Is that always related to dust and cirrus features, and therefore related to the fact that the cross-polarized signal component is missing in the case of the AEOLUS observations?

In Figure 5, there is layer from 7.5-11 km height over North/Eastern Siberia on the way to Alaska. Is that a smoke layer? Please explain and discuss. In Figure 4 (on the right and left sides of panels), there are large areas with reasonable lidar ratios around 50 sr, all over Siberia (from the ground up to the tropopause, 40N-85N, 42E-210E). Is that related to the strong Siberian fires in the summer of 2019? The smoke particles were probably spherical and the cross-polarized signal component zero…, that may explain the reasonable lidar ratios (see the MOSAiC paper of Ohneiser, ACP, 2021).

We agree that we did not have discussions about Figs. 4, 5 and only used them to show the AEL-PRO retrieval algorithm. We have added some explanations about Figs. 4, 5 close to line 208.

"The lidar ratios in Fig. 4 are often in yellow (around 100 sr). This is mostly caused by dust and/or cirrus clouds, and related to the fact that the cross-polarized signal component is missing in the Aeolus observations. The orbit shown in Fig. 4 passed over east Siberia and north America at the beginning (0-9000 km) and the end (>42000 km) of the orbit. In these areas, the large extinction coefficients (yellow to red color) and lidar ratios of 50 – 60 sr indicate that the biomass burning smoke present from the ground surface up to the tropopause. In Fig. 5 the layer from 7.5-11 km height over eastern Siberia on the way to Alaska seems like a smoke layer. There were lots of wildfires in July 2019 in east and central Sibiera, consequently biomass burning smoke was transported to north America (Johnson et al., 2021). The lidar ratio values were reasonable, most probably because of the small depolarization ratio of these smoke particles as reported by Ohneiser et al. (2021). In the southern hemisphere close to -63.2 ◦N, 228.4 ◦E the high extinction coefficient features might be due to cirrus or PSCs."

P7, section 3.1: I would mention more often that the wavelength is 355 nm.

Be clear: The CALIPSO extinction profiles are computed (estimated) from the retrieved backscatter profiles. CALIPSO cannot measure extinction profiles.

Thank you for the explanation. We added the the following sentence in sect. 3.1.

'The CALIPSO extinction coefficient profiles are computed from the retrieved backscatter profiles, not measured directly. '

We added the wavelengths in the figure captions.

An Angstroem exponent of 0.55 is almost too high for dust, may be useful for marine aerosol, but is clearly too low for other aerosols (urban haze, smoke). However, you used 0.55 for all observations, right?

I am missing a bit a discussion on the uncertainty in the CALIPSO products caused by the Angstroem exponent assumption (0.55).

Yes, we used 0.55 for all observations. We added some discussions about the uncertainty in the CALIPSO products at the end of Sect. 3.1.

'We used one one Angstroem coefficient of 0.55 to convert the CALIPSO extinction coefficient from 532 nm to 355 nm, namely the CALIPSO extinction coefficient is multiplied 1.25. If we use Asngstroem coefficient of 0.1, 0.2, or 0.9, the CALIPSO extinction coefficient will be multiplied 1.04, 1.08 or 1.44, which could cause an uncertainty of < +/-20% compared to using the Angstroem coefficient of 0.55 . '

P10, Sect 4.1.2:

P10, L290: When comparing CALIPSO and AEOLUS observations please clearly state the wavelengths of comparison. It is always 355 nm, but the reader may not remember the content of section 3.1. One could provide such information in the figure captions.

Thank you for the suggestion. We have added the wavelengths in the figure captions.

Are the AEOLUS signals stronger than CALIPSO signals because the AEOLUS wavelength is 355 nm and the CALIPSO wavelength is 532 nm?

For molecular scattering, the wavelength difference leads to substantially more molecular backscattering and extinction at 355nm compared to 532 nm. For clouds not much difference is expected. For aerosol, the differences will depend on the size distribution and composition.

P12, L357: Monthly mean extinction coefficient… for the PBL? or for the entire troposphere?

Check and update the literature (preprint status may have changed).

The monthly mean extinction coefficient profiles are for the entire troposphere.

We have modified the sentence  'The monthly mean tropospheric extinction coefficient...'  (close to line 389 in the revised manuscript).  We have updated the references.

Figure 7 and 8, the text on top of the panels is too small. Mention wavelength in the caption.

The text on the top of panels are removed and the texts are moved in the caption. We did not add wavelength in the Feature mask figures because the features do not depend on wavelength. We added the wavelengths on the figures having extinction and/or lidar ratio.

Figure 9: Mention the wavelength in the figure caption.

We have added the wavelength in caption.

Figure 10: so many lidar ratios in yellow. Is the reason discussed  in the main text body? Is that related to dust and cirrus?

The lidar ratio is an effective lidar ratio because there is only the co-polar channel. If the particles have a large depolarization ratio, the co-polar channel will have less signal than the total signal which includes both co-polar channel and cross-polar channel. So the effective S is typically larger than the real S. Cirrus and dust can have a linear depolarization ratio (depol) of 0.3, circular depolarization ratio depol_circ = 0.3*2/(1-0.3) = 0.86

beta_co = beta / (1+depol_circ )                  beta_co is backscatter coefficient of co-polar channel

S_co = S * (1+depol_circ)

= S * 1.86

We explained the effective S in sect. 2.3 close to line 187

S in Figure 10 is explained in Sect. 4.1.2 line 327 - 333.

"Fig. 10(d) shows lots of large S values (in yellow color), which are mostly related to cirrus clouds and dust. We have analysed the distribution of S for dust at latitude 15–30 ◦N and height bins below 5 km, and for smoke at latitude 10–25 ◦S, height bins below 5 km. As shown in Fig. 11, the lidar ratios for the smoke and dust scenes have different distributions. The smoke lidar ratio has a peak close to 75 (72–78) sr but the distribution is rather broad from 25 to 120 sr. The dust lidar ratio has a large peak close to 54 (48-60) sr and a second peak close to 102 sr. In AEL-PRO the a priori of aerosol lidar ratio is 60 sr, depolarization 0.15. The retrievals are not sensitive to the a priori values. Floutsi et al. (2023) reported that the Saharan dust has the S of 53.5±7.7 sr and the depolarization ratio of 0.244±0.025, the effective S can be 1.646 times of the true S. The smoke has the S of 68.2±7.4 sr and the depolarization ratio of 0.027±0.013, the effective S can be 1.055 times of the true S. We think the lidar ratios in Fig. 11 are reasonable compared to the Sahara dust and smoke lidar ratios."

Figure 11: the lidar ratio distribution belongs to what height range? .

All altitude range below tropopause. We have replaced Fig. 11 because reviewer 1 found it confusing.

Figure 12: Mention the wavelength in the figure caption.

We have added the wavelength in the caption.

Figure 13: What is the message of the correlation? What is the impact of the assumed Angstroem exponent of 0.55?

A: The correction is small (presumably due to measurement error and real variability) but no large offset exists.

Based on DeLiAn paper, the Angström is 0.1+/0.2 for Sahara dust, 0.2+/- 0.1 for central Asian dust, 0.1+/- 0.1 for middle eastern dust, dust and marine 0.5+/- 0.5. Dust and pollution 0.7+/- 0.4.

The area we used in the analysis is main Sahara dust and dust and marine when dust over ocean.

So we used is close to dust and marine, if we use 0.1, 0.2, the CALIPSO ext at 355 nm would be smaller. The conversion factor is 1.25 to 1.08, 1.04.

We added the discussion in the revised manuscript in sect. 3.1

"We used one Angström coefficient of 0.55 to convert the CALIPSO extinction coefficient from 532 nm to 355 nm, namely the CALIPSO extinction coefficient is multiplied by 1.25. If we use an

Angström coefficient of 0.1, 0.2, or 0.9, the CALIPSO extinction coefficient will be multiplied by 1.04, 1.08 or 1.44, which could cause an uncertainty of <±20% compared to using the Angström coefficient of 0.55."

Figure 14: The extinction values from 5-15 km are confusing. The tropopause for latitudes from 0-30N is probably around 15-17 km height. So, probably only tropospheric extinction profiles are shown? Why is the agreement of the different extinction profiles so bad for heights >5km? The background extinction coefficients in the clean upper troposphere (and lower stratosphere) should be around 0.75-1 Mm-1 at 355 nm and about 0.25-0.5 Mm-1 at 532 nm. Is again an Angstroem exponent of 0.55 used in the conversion of the CALIPSO data?

We only used the tropospheric extinction coefficients. Angström exponent  is 0.55 for all conversions.

Figures 15 and 16. Do we need these figures?

We will keep the figures. They show the AOT map and the orbits and provide useful context

---

## Author Comment (AC2)

We appreciate referee #1 for the comments and suggestions. We have answered all questions and revised the manuscript. We also modified figures 4, 14, 15, 16, so that the AEL-PRO figures come first, then the SCA-mid or CALIPSO figures.  In this document, the answers are given in blue text.

**Review report egusphere-2024-731**

The authors present an assessment study of the Aeolus' feature mask and vertical extinction profiles versus CALIPSO data. Aeolus products have been obtained via the implementation of the AEL-FM and AEL-PRO retrieval algorithms adapted from the A-FM and A-PRO algorithms, which have been developed for the EarthCARE HSRL ATLID. The analysis focuses on dust-rich scenes probed by the two spaceborne instruments across N. Africa in October 2018 and May 2019. The study is well-organized, and all the essential details are well presented and discussed. Therefore, I recommend that the manuscript be published after addressing the minor comments provided below.

1. **Lines 64-65:** To what extent your results will be affected by the consideration of L1B data generated with the most recent Baseline version (i.e., Baseline 16)?

We added some answers in Sect. 3.2 Aeolus data.
"The AEL-FM, AEL-PRO algorithms derive the attenuated Rayleigh and Mie backscatter signals from the L1B Mie measurement data (counts), the impacts from L1B and auxiliary data are mainly the hotpixel detection, dark current, background signal. We do not expect significant changes in these parameters between L1B baselines 14 and 16."

2. **Lines 91-92:** Is there any threshold on the number of counts?

I am not sure about the question. There is no threshold on the number of counts when separating the Mie and Rayleigh signals. The ACCD pixels from 6 to 11 are used to calculate Mie channel signal (not counting the leading 2 pixels).

3. **Line 177:** Which is the source of the a priori lidar ratios and the particle effective area radii?

We do not have a specific reference for the a priori lidar ratios and particle effective area radii but we try to choose some reasonable values as in the DeLiAn paper and in the simulated data for ATLID.

We added the follow sentence  in the manuscript close to line 174. We also exchanged the order of two paragraphs starting at lines 171 and 179.
"In a configuration file, the a priori values of lidar ratio (S) and Ra are specified for water clouds, ice clouds, two kinds of stratospheric ice clouds, aerosols and stratospheric aerosols according to the values in the simulated data for ATLID (Donovan et al., 2023) and in Floutsi et al. (2023)."

Floutsi, A. A., Baars, H., Engelmann, R., Althausen, D., Ansmann, A., Bohlmann, S., Heese, B., Hofer, J., Kanitz, T., Haarig, M., Ohneiser, K., Radenz, M., Seifert, P., Skupin, A., Yin, Z., Abdullaev, S. F., Komppula, M., Filioglou, M., Giannakaki, E., Stachlewska, I. S., Janicka, L., Bortoli, D., Marinou, E., Amiridis, V., Gialitaki, A., Mamouri, R.-E., Barja, B., and Wandinger, U.: DeLiAn – a growing collection of depolarization ratio, lidar ratio and Ångström exponent for different aerosol types and mixtures from ground-based lidar observations, Atmos. Meas. Tech., 16, 2353–2379, https://doi.org/10.5194/amt-16-2353-2023, 2023.

4. **Lines 179-181:** Can you please explain better this sentence? What do you mean "… *otherwise the lidar-ratio supplied by the classification procedure is used.*"?

The sentence has been rewritten as, 'However, the Pass-I output is only valid for the weak features, for the strong features or invalid Pass-I output, Pass-II selects the a priori values from the configuration file. ' close to line 184.

5. **Lines 234-236:** In my opinion it would be quite interesting to show this comparison and briefly discuss the obtained outcomes. It is well known that when non-spherical particles (e.g., dust) are probed by ALADIN it is expected a "weak" performance in terms of reproducing the backscatter coefficient (for reasons already stated in the manuscript). Taking into account that there is a sufficient volume of Aeolus-CALIPSO collocated data, a better assessment can be given than those in Abril-Gago et al. (2022) and Gkikas et al. (2023), who presented single (few) dust cases.

Thank you for the suggestion and references. We prefer to keep the comparison focus on extinction coefficients in this paper.
If we compare the backscatter coefficients, we would need to assume/choose a depolarization ratio and do some simulations using different particles to understand our findings. This is out of scope for this current paper. We could write another paper about the evaluation of the backscatter coefficients.

6. **Line 286:** I would suggest to remove this sentence since CALIPSO assigns a lidar ratio for each aerosol type.
I cannot find the sentence related to CALIPSO close to Line 286. We removed the sentence, 'We do not compare the lidar ratio with CALIPSO data in this paper'.

7. **Lines 294-300:** I am confused with this part of the text. Why are you considering all aerosol subtypes across the scene in order to reproduce the frequency histogram of S values? Do you think that it would be better to reproduce the histograms for specific aerosol types (dust and smoke for this case)? I think that the dust lidar ratio given by Song et al. (2023) are substantially higher than those provided in the DeLiAn database (Floutsi et al., 2023).

The histogram was used to show the values in the AEL-PRO for all cases, including clouds and aerosols.

In the AEL-PRO product, we do not have aerosol subtypes, so I did not separate aerosol subtypes when making the histogram of S. I thought there would be a bi-mode distribution if there were different S for dust and smoke burning but it did not show up.

We have checked the distribution of S for dust (lat 15-30 N, below 5 km) and smoke (lat 10-25 S, below 5km). We can see the lidar ratio for the smoke case is on average larger than that for the dust case.

In AEL-PRO the a priori of aerosol lidar ratio is 60, depolarization 0.15.

In the figure we find the smoke lidar ratio has a peak close to 75  (72-78) sr but the distribution is rather broad from 25 to 120 sr.

The dust lidar ratio has a large peak close to 54 (48-60), a second peak close to 102. The lidar ratios are reasonable compared to the Sahara dust and smoke lidar ratio in the DeLiAn data set.

In the DeLiAn paper,
Saharan dust S = 53.5 +/- 7.7 sr    depol = 0.244 +/- 0.025
Smoke          S = 68.2 +/- 7.4 sr    depol = 0.027 +/- 0.013

[Figure]

Fig 11

We replaced the S distribution figure with this new figure and add explanations in the revised manuscript between lines 323 – 333.
In lines 187 – 195 we now explain the effective lidar ratio (related to the fact that Aeolus uses circular depolarization and only measures the co-polar backscatter).

8.  **Line 304:** What do you mean with the term "error" for the CALIPSO extinction coefficients?
The error for CALIPSO extincton coefficients is Extinction_Coefficient_Uncertainty_532 in CALIPSO L2 data.  We have added the explanation and changed the error to uncertainty in the manuscript.

9. **Section 4.2:** Do you see any noticeable differences between daytime and nighttime conditions?

We did not look at the daytime and nighttime data separately because we did not have lots of collocated data to separate to two datasets.
Recently we looked at the AEL-PRO data for in June 2020 over Sahara desert and found the afternoon orbits have a slightly larger extinction coefficients than the morning orbits. However, this analysis is not relevant for this paper.

10. **Section 4.1:** Are you taking into account all the CALIPSO retrievals or are you processing only those tagged as "dust" in the classification scheme?

We used CALIPSO data for all cases, not only the cases tagged as "dust'. The dust aerosols are selected in the region of lat [0-30] °N, lon [-10 – 50] °E, where most aerosols were dust but there could be other aerosol types, like sea salt.

11. **Lines 335-336:** How much your results would be affected in the case of using more realistic aerosol speciated lidar ratios (see DeLiAn)?

The lidar ratio is a retrieved parameter, we have tried to use realistic a priori values for the lidar ratio. However, the retrieved lidar ratios are effective ones. They are appropriate to co-polar backscatter only (since ALADIN only has a co-polar channel and no cross-polar backscatter is detected). The difference between this effective S and the "normal" S depends on the particle circular depolarization ratio and smaller for particles having small depolarization ratios. The retrieval algorithm is not very sensitive to the a priori S value. We have discussed this issue  close to lines 187-195..

**Lines 364-365:** It would be nice to provide further explanation regarding this assertion, highlighting the necessity of the deployment of a cross-polar channel on the Aeolus-2 satellite mission.

The cross-polar channel is needed to derive the depolarization ratio and lidar ratio appropriate for total backscatter. These two parameters are essential in separate clouds, aerosols and subtype aerosols. We have added some discussion in Sect. Conclusion and outlook.